# *Diphyllobothrium* sp. and Other Parasites of Migrating and Rare Fish Species in the Southern Baltic Sea and Coastal Waters, Poland

**DOI:** 10.3390/ani14071029

**Published:** 2024-03-28

**Authors:** Izabella Rząd, Beata Więcaszek, Angelika Linowska, Agata Korzelecka-Orkisz, Ewa Dzika

**Affiliations:** 1Institute of Marine and Environmental Sciences, University of Szczecin, Wąska 13, 71-415 Szczecin, Poland; 2Department of Hydrobiology, Ichthyology and Reproductive Biotechnology, West Pomeranian University of Technology in Szczecin, Kazimierza Królewicza 4, 71-550 Szczecin, Poland; beata.wiecaszek@zut.edu.pl (B.W.); angelika.linowska@zut.edu.pl (A.L.); akorzelecka@zut.edu.pl (A.K.-O.); 3Department of Medical Biology, School of Public Health, University of Warmia and Mazury in Olsztyn, Zołnierska 14c, 10-561 Olsztyn, Poland; e.dzika@uwm.edu.pl

**Keywords:** Baltic Sea, *Diphyllobothrium*, fish, parasites, one health

## Abstract

**Simple Summary:**

The aim of this study was to determine the occurrence of parasites in migrating and rare fish in the Pomeranian Bay and the Szczecin Lagoon and to determine the composition of the diet of these fish. The fish for analysis, i.e., saithe, European hake, dab, common sole, whiting, tub gurnard, surmullet, Atlantic mackerel, brill, and plaice, were obtained in the years 2010–2019. This study is the first to analyse the parasite fauna of sole, dab, hake, whiting, and plaice in the Pomeranian Bay, as well as saithe from the Szczecin Lagoon (Poland). The presence of parasites was confirmed: two species of protozoa; the larva of the tapeworm *Diphyllobothrium* sp.; seven species of nematodes, including the typical marine nematode *Capillaria (Procapillaria*) *gracilis*, rarely recorded in Poland; and one acanthocephalan. Analysis of the contents of the stomach and intestines showed that most of the fish had empty stomachs. Because the fish species analysed in the study are not typically present in the Baltic and we do not know how long they feed while they are in the Baltic, we cannot be certain which parasites they acquired in the water bodies analysed in the study and which were introduced during the migration of the fish. The round goby, an invasive fish species, was recorded for the first time in the diet of sole, and the freshwater tench was recorded for the first time in the diet of saithe.

**Abstract:**

The present study is the first to analyse the parasite fauna of sole *Solea solea*, dab *Limanda limanda*, hake *Merluccius merluccius*, whiting *Merlangius merlangus*, and plaice *Pleuronectes platessa* in the Pomeranian Bay, as well as saithe *Pollachius virens* from the Szczecin Lagoon (Poland). The aim of this study was to determine the occurrence of parasites in migrating and rare fish in the Pomeranian Bay and the Szczecin Lagoon and to determine the composition of the diet of these fish. The fish for analysis were obtained in the years 2010–2019. The typical marine nematode *Capillaria (Procapillaria*) *gracilis*, rarely recorded in Poland, was found, in addition to the following parasites: *Ichthyophonus hoferi*, *Trichodina jadranica*, *Diphyllobothrium* sp., *Dichelyne* (*Cucullanellus*) *minutus*, *Raphidascaris acus*, *Anisakis simplex*, *Contracaecum osculatum*, *Hysterothylacium aduncum*, *Pseudoterranova decipiens*, and *Echinorhynchus gadi*. Because the fish species analysed in the study are not typically present in the Baltic (with the exception of plaice), and because we do not know how long they feed while they are in the Baltic, we cannot be certain which parasites they acquired in the water bodies analysed in the study and which were introduced during the migration of fish. Although these fish are outside of their normal geographic range (except for plaice), in the new environment, there were enough suitable intermediate hosts for the parasites of these fish to complete their life cycle and survive.

## 1. Introduction

The Pomeranian Bay (in the Bornholm Basin, SW Baltic Sea) is a region with riverine water input and water exchange with adjacent open seawaters [1]. The Pomeranian Bay connects with the Szczecin Lagoon via three outlets [2]. It is one of the most important ecological areas in the Baltic Sea. Sandbanks, banks of blue mussels, and reefs are sensitive habitats with their adapted, species-rich communities, and they are highly representative of the Baltic Sea. The Pomeranian Bay is also of great economic importance for fisheries, with commercial catches of Atlantic cod *Gadus morhua* L., 1758, sprat *Sprattus sprattus* (L., 1758), Baltic herring *Clupea harengus* L., 1758, and European flounder *Platichthys flesus* (L., 1758) [3]. The periodic occurrence of non-native and very rare species of fish in the Pomeranian Bay is most likely associated with inflows of saline water from the North Sea and/or climate change [4]. Parasitological studies of fish caught for commercial purposes are particularly important due to the potential risk associated with zoonotic parasites, i.e., those that can infect people. Examples include nematodes of the family Anisakidae and some cestodes of the family Diphyllobothriidae. Parasites of commercially caught fish have been thoroughly investigated due to their direct impact on human health. Parasites of rare or migrating fish, however, are little known. There is a gap in knowledge regarding parasite acquisition during the migration of fish and the consequences of the transport of parasites between ecosystems by rare and migrating fish for consumer safety and ecosystem health.

Several groups of rare or migrating fish species can be distinguished based on the literature on the Pomeranian Bay. The first group includes occasional marine visitors that migrate seasonally from the open northeastern Atlantic, the North Sea, and the Kattegat to the Baltic but are unable to form self-sustaining populations there [5], e.g., tub gurnard *Chelidonichthys lucerna* (L., 1758) and Atlantic mackerel *Scomber scombrus* L., 1758 [6], a long-migrating species. Other marine fish species from the North Sea that periodically migrate to the Baltic Sea include whiting *Merlangius merlangus* (L., 1758), saithe *Pollachius virens* (L., 1758), seabass *Dicentrarchus labrax* (L., 1758), European anchovy *Engraulis encrasicolus* (L., 1758), mullets *Chelon ramada* (Risso, 1827) and *Chelon labrosus* (Risso, 1827), jack-horse mackerel *Trachurus trachurus* (L., 1758), common sole *Solea solea* (L., 1758), and swordfish *Xiphias gladius* L., 1758 [7]. A second group of fish species is expected to extend its distribution northward (from the southern North Sea towards the southwestern Baltic Sea) in response to climate warming, including surmullet *Mullus surmuletus* L., 1758 [8] and probably representatives of the genus *Chelon* [9]. The third group of species is typically fish overflow from the Kattegat and the Belt Sea, such as European hake *Merluccius merluccius* (L., 1758) [10], the deep-water shark *Etmopterus spinax* (L., 1758) [11], ballan wrasse *Labrus bergylta* Ascanius, 1767 [12], and greater weever *Trachinus draco* L., 1758 [7].

The literature on the parasite fauna of fish species from the Pomeranian Bay and Szczecin Lagoon is scarce. Data on the parasite fauna of rare and endangered fish species can be found in papers by Więcaszek et al. [13,14,15], while parasites of commercial fish species are described by Sobecka et al. [16] (Atlantic cod), Korlatowicz and Piasecki [17] (European flounder), Radačovska et al. [18] (perch *Perca fluviatilis* L., 1758), Bielat et al. [19] (perch and pike-perch *Sander lucioperca* (L., 1758)), and Sobecka and Słomińska [20] (perch and bream *Abramis brama* (L., 1758)).

Parasite fauna of non-commercial but biologically important fish species are described in the works of Sobecka and Łuczak [21] (viviparous eelpout *Zoarces viviparus* (L., 1758)), Grabda [22] (garfish *Belone belone* (L., 1761)), and Pilecka-Rapacz et al. [23] (smelt *Osmerus eperlanus* (L., 1758)). To date, the following parasites have been recorded in rare fish species examined in the Pomeranian Bay: *Epistylis colisarum* (Foissner and Schubert, 1977) (Ciliophora, Epistylididae), and *Chilodonella hexasticha* (Kiernik, 1909) (Ciliophora, Chilodonellidae) in thicklip mullet; *Contracaecum osculatum* (Rudolphi, 1802) (Nematoda, Anisakidae) in jack-horse mackerel and tub gurnard; *Pseudoterranova decipiens* (Krabbe, 1878) (Nematoda, Anisakidae) in surmullet and in tub gurnard; *Hysterothylacium aduncum* (Rudolphi, 1802) (Nematoda, Raphidascarididae) and *Corynosoma strumosum* (Rudolphi, 1802) (Acantocephala, Polymorphidae) in tub gurnard; *Pomphorhynchus laevis* (Zoega in Müller, 1776) (Acantocephala, Pomphorhynchidae) in surmullet; and *Unio* sp. (Mollusca, Unionidae) in thicklip mullet [13]. Więcaszek et al. [14] recorded larvae (L3) of *Contracaecum osculatum*, *Raphidascaris acus* (Bloch 1779) (Nematoda, Raphidascarididae), *Podocotyle reflexa* (Creplin 1825) (Platyheminthes, Opecoelidae), and *Ichthyophonus hoferi* Plehn and Mulsow, 1911 (Choanozoa, Ichthyophonidae) in lumpfish *Cyclopterus lumpus* L., 1758, and *Echinorhynchus gadi* Zoega in Müller, 1776 (Acantocephala, Echinorhynchidae) in fourbeard rock ling *Enchelyopus cimbrius* (L., 1766). Więcaszek et al. [15] identified *Myxobolus musculi* Keysseliz, 1908 (Cnidaria, Myxobolidae), *Pomphorhynchus laevis* and *Rhabdochona hellichi* (Śrámek, 1901) (Nematoda, Rhabdochonidae) in barbel *Barbus barbus* (L., 1758), and *Trichodina jadranica* Raabe, 1958 (Ciliophora, Trichodinidae) in brill *Scophthalmus rhombus* (L., 1758).

The aim of this study was to determine the occurrence of parasites in migrating and rare fish in the Pomeranian Bay and the Szczecin Lagoon and to determine the composition of the diet of these fish. This study is the first to analyse the parasite fauna of sole, dab *Limanda limanda* (L., 1758), hake, whiting, and plaice *Pleuronectes platessa* L., 1758 in the Pomeranian Bay, as well as saithe from the Szczecin Lagoon.

## 2. Material and Methods

### 2.1. Study Area

The parasites for the study were collected from fish caught as bycatch during monitoring surveys of commercial fish species in the Pomeranian Bay (Bornholm Basin), from May to December (2010–2019). The fishing was conducted from the research vessel at depths of 9.8–14.6 m, along routes at 53°57′ N–54°10′ N; 14°19′ E–14°47′ E. The saithe specimen was caught in the northern part of the Szczecin Lagoon (53°48′ N; 14°22′ E).

### 2.2. Ichthyological Analysis

The fish species from which parasites were collected for analysis are very rare in the Pomeranian Bay. They had probably migrated from the North Sea [4]. Twenty fish specimens belonging to 10 species were examined in the study. Representatives of European hake, dab and common sole were recorded for the first time in the Pomeranian Bay, while saithe was noted for the first time in the Szczecin Lagoon (connected with the Pomeranian Bay) [4]. Whiting, tub gurnard, surmullet, Atlantic mackerel, and brill had previously been found in the Pomeranian Bay, but very rarely. Plaice had recently been recorded in the Pomeranian Bay, but rarely [24] (Table 1). The identification of fish species was based on partial sequences of the cytochrome c oxidase subunit I (*COI*), cytochrome b (*cytb*), and rhodopsin (*rho*) genes and morphometric diagnostic characters. A detailed description is found in Więcaszek et al. [4].

### 2.3. Parasitological Study

Parasitological necropsies of the fish were performed in accordance with standard procedures [25,26,27,28,29,30]. Parasitological examination focused on the skin, vitreous humour, eye lens, oral and nasal cavities, gills, gonads, gastrointestinal tract, kidneys, swim bladder, urinary bladder, gall bladder, peritoneum, and muscles. The parasites found were prepared for species determination by viewing them in transient light, immersed in glycerine or preserved in 70% ethanol. The morphology of the parasites was examined using a stereoscopic microscope (5–40×) and a biological light microscope (10–400×). Molecular methods were necessary to identify the parasite larva from the cyst found in the saithe (see below). The remaining species were identified on the basis of morphological characters. Species were identified using keys and original works [29,31,32,33].

The initial research questions and hypotheses concerned the possibility of fish transporting parasites from other ecosystems that could pose a threat to native fish in the Pomeranian Bay. Another interesting question was whether we would find parasites known in fish in the Pomeranian Bay but previously unknown in these alien fish species.

### 2.4. Gut Content Analysis

Fish were dissected ventrally, and each prey item in their gastrointestinal tracts was identified to the lowest possible taxonomic level with a Nikon SMZ1000 microscope coupled with ToupViev 3.7 image analysis software (Nikon, Zürich, Switzerland). When fish species were recorded in the diets of predators, the prey fish were measured, and, if possible, otoliths were collected to confirm identification. The identification keys used for invertebrates were those suggested by Fish and Fish [34] and Hayward and Ryland [35].

### 2.5. DNA Extraction, PCR Assay, and Sequence Analysis

DNA from one of the cestode cysts found in saithe was extracted using the High Pure PCR Template Preparation Kit (Roche, Berlin, Germany). The DNA passed a quality and quantity check involving separation in a 1.5% agarose gel (no signs of DNA degradation) and spectrophotometric measurement on the NanoDrop 2000 (Thermo Scientific, Altrincham, UK). Primers (ZX-1 5′-ACCCGCTGAATTTAAGCATAT-3′ and 1200-R 5′-GCATAGTTCACCATCTTTCGG-3′) from Caira et al. [36] were used for the amplification of a partial lsrDNA fragment. Amplifications were conducted on a T100™ Thermal Cycler (Bio-Rad, Feldkirchen, Germany) using the GoTaq PCR kit (Promega, Walldorf, Germany), including 5 μL of Green GoTaq^®^ Flexi Buffer, 2.5 μL of MgCl_2_ (25 mM solution), 0.5 μL of PCR Nucleotide Mix (10 mM), 0.125 μL of GoTaq^®^ DNA Polymerase (5 µL^−1^), 0.5 μM of each primer, and 5 μL of DNA template in a final volume of 25 μL. The results of all PCR reactions were assessed by 2% gel electrophoresis and subsequently bidirectionally sequenced by Genomed (Warsaw, Poland). The raw reads of PCR products were assembled using Geneious 8.0 [37] and compared against GenBank sequences using BLAST [38] to identify the sequences with the highest similarity. Next, the sequences were aligned to calculate the percentage similarity between them, and a phylogenetic tree was constructed under the maximum likelihood (ML) criterion (general time-reversible GTR model, 10,000 bootstrap) using the PhyML 3.0 plugin in Geneious 11.1.5 [37,39].

The specimens of the mussel *Mytilus* sp. identified in the stomach of the plaice were highly digested, so species determination by molecular methods was not possible. Therefore, additional samples of mussels were collected from the seafloor in the area where the plaice was acquired (53°57′05″ N 14°20′22″ E). Based on morphological comparisons of mussel samples from the stomach and the seafloor, the specimens were assigned to the genus *Mytilus*. Subsequently, to determine the species identity of mussels from the seafloor, DNA from those samples was extracted and assessed as described above. PCR and downstream sequence processing and comparisons were performed according to Zbawicka et al. [40].

## 3. Results

### 3.1. Parasites

Parasites were detected in eight species of fish. The parasites included Protozoa (two species), Cestoda (one genus), Nematoda (seven species), and Acanthocephala (one species). In the saithe intestine, the typical marine cod nematode *Capillaria* (*Procapillaria*) *gracilis* (Bellingham, 1840), sporadically recorded in fish caught in Poland, was identified (Table 2).

Among parasitic Protozoa, representatives of both ectoparasites—*Trichodina jadranica*—and endoparasites—*Ichthyophonus hoferi*—were identified. In European hake and tub gurnard, mass infection by *I. hoferi* was observed, while in the remaining species, Protozoa occurred sporadically.

In the saithe, two milky-white cysts with cestode plerocercoids [cyst size 1023 µm × 958 µm] were located on the surface of the stomach. Based on analysis of the partial sequence of lsrDNA (688 bp) from one of the cysts, it was identified as *Diphyllobothrium* sp. (Metazoa, Diphyllobothriidae) (Figure 1). The taxonomy of Diphyllobothriidae is controversial; we used the name *Diphyllobothrium* (not *Dibothriocephalus*), as recommended by Scholz et al. [41]. Cysts were located in the stomach of the fish. The location of *Diphyllobothrium* sp. in the stomach of fish is not typical; this may be a stage in the life cycle of the parasite in which it was found before it could penetrate the stomach wall to the body cavity. This question is discussed at greater length in the Section 4.

Nematodes were the most numerous and also the most species-rich group of fish parasites from the Pomeranian Bay. The most frequently recorded helminth was the nematode *Dichelyne* (*Cucullanellus*) *minutus* (Rudolphi, 1819), found in the gut of the European plaice. Parasitological analysis of fish confirmed the presence of Anisakidae nematode larvae, including invasive third-stage larvae of *Anisakis simplex* (Rudolphi, 1809) isolated from the liver of tub gurnard, the peritoneum and gonads of Atlantic mackerel, and the intestine and gonads of dab. In tub gunnard, *Contracaecum osculatum* was found on the pyloric caeca and in the stomach, and *Pseudoterranova decipiens* was found on the pyloric caeca. *Hysterothylacium aduncum* and *Raphidascaris acus* were found in the intestine of fish (Table 2).

One of the most frequently recorded Acanthocephalan species in Poland, *Echinorhynchus gadi*, was located on the surface of the dab intestine and in the intestinal lumen in the saithe. The number of specimens of this parasite did not exceed three in either fish (Table 2). Among the 10 fish species examined, 2 (whiting and common sole) were entirely free of parasites.

### 3.2. Results of Gut Content Analysis

The stomach contents of the fish specimens are presented in Table 3. Most of the fish (13) had empty stomachs, including all specimens of European hake, surmullet, dab, and Atlantic mackerel and single specimens of brill and common sole. Specimens of the invasive round goby *Neogobius melanostomus* (Pallas, 1814) were noted in the stomachs of two fish specimens (common sole and brill). The remains of caridean shrimp *Crangon crangon* (L., 1758) (Decapoda, Malacostraca) and three skeletons of European flounder (Pleuronectiformes) were recorded in one of the five specimens of tub gurnard, while the remains of Mysidacea (Mysida, Malacostraca) were noted in another tub gurnard specimen. One specimen of *C. crangon* was recorded in the stomach of whiting, and scales of freshwater tench, *Tinca tinca* (L., 1758) (Cypriniformes), were noted in the stomach of saithe. Numerous remains of the bivalve *Mytilus* sp. (Bivalvia, Mytilida) and unidentified fish roe were noted in the plaice specimen. Based on morphological features, mussel samples collected in the same area as fish samples, including those recorded in the stomach of plaice, were assigned to the genus *Mytilus*. Subsequently, molecular analyses performed using two species-specific single nucleotide polymorphisms (SNPs) for the genus *Mytilus* revealed that the samples collected from the sea floor were *Mytilus edulis* L., 1758. It is likely that the mussels from the stomach of plaice belonged to this species as well.

## 4. Discussion

The fish in which parasites were collected for analysis are very rare species in the Pomeranian Bay, including some that were found for the first time in this region (European hake, common sole, and dab). Their appearance is most likely linked to the extension of the ranges of species due to climate warming or the inflow of saltwater from the North Sea [4]. Several groups of fish species from which parasites were collected for analysis can be distinguished. The first group comprises occasional marine visitors that seasonally migrate from the open Northeast Atlantic, the North Sea, and the Kattegat to the Baltic but are unable to establish self-sustaining populations there, such as tub gurnard. They are sporadically distributed in the western Baltic Sea. Another example is the mackerel, a species with long migratory patterns, whose migration to and from the Baltic is influenced in part by climatic variables. A second group of fish species is anticipated to expand its distribution northward in response to climate warming; the surmullet belongs to this group [8]. The third group of species comprises fish that overflow from the Kattegat and the Belt Sea, including European hake, saithe, and whiting, abundant in the northern North Sea but absent from the western Baltic Sea [10]. The remaining fish species are considered native to the Baltic but not to the Pomeranian Bay (except for plaice). According to Heessen et al. [10], brill is regularly observed off Cape Arkona and around Bornholm; however, there were no landings there in the years 2012–2016. Its presence in the Pomeranian Bay could result from either active migration from these areas or passive translocation with inflows of waters from the western Baltic. The first record of brill in the Pomeranian Bay was in 2014 [15]. Sole, as per Heessen et al. [10], is a southern species commonly found in the Kattegat (salinity > 25), but seldom encountered in the Baltic. Dab currently resides in the Kattegat and the western Baltic, whereas in the central Baltic, the stock experienced a collapse after the Second World War and never recovered due to unfavourable hydrographic conditions and predation by cod [5].

The species richness of parasites consisted of Protozoa and helminths, including cestodes, nematodes, and acanthocephalans, while no trematodes were recorded (Monogenea or Digenea).

### 4.1. Helminths

An important discovery is the presence of the typical marine nematode *Capillaria* (*Procapillaria*) *gracilis* (found in saithe) [42]. Sobecka [43] found numerous specimens of *Capillaria gracilis* (synonym) in cod from Øresund (a high prevalence of 66% and a high intensity of infection) and one specimen of this nematode in cod from the Irminger Sea. According to Sobecka, a factor limiting the spread of this nematode in the Baltic is low salinity [43]. Despite the fact that suitable intermediate hosts for this nematode are available in the Baltic Proper and the eastern Baltic, its presence has not previously been recorded [43]. On the Polish coast of the Baltic Sea, Sobecka and Łuczak [21] recorded the presence of this nematode in eelpout in the Oder River estuary (at the mouth of the river Dziwna). Pilecka-Rapacz and Sobecka report the occurrence of *Capillaria gracilis* in cod caught in the southern Baltic [44]. Klimpel et al. [45] recorded the presence of this nematode in surmullet in the North Sea (prevalence of 7.7%) [45]. This nematode lives in the mucosa of the posterior part of the intestine in marine fish, mainly Gadidae, such as European hake, Atlantic cod, poor cod *Trisopterus minutus* (L., 1758), and whiting pout *T. luscus* (L., 1757), as well as in fish of the genus *Myoxocephalus* sp. and others [33]. The intermediate host is the oligochaete *Tubifex costatus*; Chironomid larvae are alternative intermediate hosts. The role of the paratenic host can be played by numerous invertebrates [33,42]. The remaining parasites found in our study had previously been noted in the Baltic more often than *C. gracilis* in various species of fish [46]. These parasites can potentially complete their life cycle in the Baltic, as their presence (both larvae and adult forms) has been recorded in suitable intermediate, paratenic and definitive hosts in the Baltic.

The helminths included parasites that could potentially pose a threat to humans: the cestode *Diphyllobothrium* sp. and the nematodes *Anisakis simplex*, *Pseudoterranova decipiens* and *Contracaecum osculatum*. However, for the purpose of assessing the zoonotic risk posed by these parasites, parasitological analyses should always take into account where they are located in the fish. These parasites were found in parts of the fish that are usually not consumed: in the stomach (*Diphyllobothrium* sp.), liver, peritoneum, and intestine (*A. simplex*); the pyloric caeca (*P. decipiens*); and the pyloric caeca and the stomach (*C. osculatum*). The results of our study suggest that *A. simplex* found in Atlantic mackerel and dab may pose a risk to people eating the gonads of fish (Table 2). In the case of *Diphyllobothrium* sp., without knowing which species we are dealing with, we cannot estimate the risk of infection in humans because not all species of this genus pose a zoonotic risk. The remaining species of helminths and Protozoa have varying degrees of pathogenicity for fish and other hosts in their life cycles.

Because the fish examined in the present study are rare in the Baltic and our sample of fish was small, it is difficult to assess their impact on other fish populations and the ecosystem. On the one hand, parasites are a natural component of every ecosystem [47]. On the other hand, migrating fish can contribute to the translocation of alien parasite species between ecosystems, e.g., [48]. This information should be taken into account when implementing the One Health concept [49].

Saithe, a marine fish species in which we found the cestode *Diphyllobothrium* sp., plerocercoid, was caught in the freshwater Szczecin Lagoon, connected to the Pomeranian Bay. Since only the scales of tench were found in the stomach of the saithe, it is likely that the source of the *Diphyllobothrium* sp. larvae was fish of another species previously ingested and digested, and that the larvae had not yet migrated from the stomach to the muscles of saithe. *Diphyllobothrium* sp. larvae have previously been noted in the stomach of cod [50]. The last 20 years have seen significant advances in the knowledge of cestodes of the family Diphyllobothriidae (order Diphyllobothriidea). A breakthrough in the study of Diphyllobothriidea was their separation from the order Pseudophyllidea and recognition as a separate order by Kuchta et al. [51]. Later, Waeschenbach et al. [52] performed the most comprehensive phylogenetic analysis of Diphyllobothriidea to date using DNA sequences from verified and confirmed specimens of these cestodes, showing that this may be an important means of reliably identifying pathogenic species. In addition, they solved taxonomic problems of human-infecting species [52].

*Diphyllobothrium* cestodes are the largest cestodes recorded in humans and are the cause of the parasitic disease diphyllobothriosis. In the near future, we can expect the expansion of diphyllobothriosis transmitted by fish due to the increase in risk factors, such as the growing popularity of consumption of raw fish in dishes such as sushi, international travel, and human migration, as pointed out by Kuchta et al. [53], Broglie and Kapel [54], and Arizono et al. [55]. Valuable studies summarize information about the best-known cestode species transmitted by fish, such as *Diphyllobothrium latum* (Linnaeus, 1758), *D. nihonkaiense* Yamane, Kamo, Bylund and Wikgren, 1986, and *D. dendriticum* (Nitzsch, 1824), and information about the occurrence of rarely recorded species that can infect humans [56,57]. People become infected with cestodes *Diphyllobothrium* by eating live larvae (plerocercoids) of cestodes located in the tissues of fish that have not undergone procedures (cooking or freezing) to kill them. Infection may take place through home-prepared caviar, e.g., from the roe of pike, salmon, and other fish. *Diphyllobothrium* cestodes are cosmopolitan and occur mainly endemically in regions where culinary habits include the consumption of raw fish. In Poland, they have been noted in lake fish [46] and in fish in the Baltic [58]. These cestodes have also been found in foxes [46].

Among the 14 genera of diphyllobothriid tapeworms, which include 58 species, 10 develop in marine habitats and 4 develop in terrestrial habitats—in mammals (Mammalia, 12 cestode genera) and birds (Aves, two cestode genera) [41]. The vast majority of cestode species of the family Diphyllobothriidae are parasites of wild animals, and their definitive hosts are mainly marine and terrestrial mammals [41]. People mainly become infected by species of the genera *Dibothriocephalus* and *Spirometra*. According to Scholz and Kuchta [56], who conducted a review of reports of cestodes in humans (excluding *Spirometra*), the cestodes *Adenocephalus pacificus* Nybelin, 1931, *Dibothriocephalus dendriticus*, *D. latus*, *D. nihonkaiensis*, *Diphyllobothrium balaenopterae* (Lönnberg, 1892), and *D. stemmacephalum* Cobbold, 1858 are species that can be regarded as human parasites [56].

For example, the cestode previously belonging to the genus Diphyllobothrium—*Diphyllobothrium latum*, with the currently accepted name of *Dibothriocephalus latus* (Linnaeus, 1758) [59]—is frequently mentioned in the literature. The definitive hosts of *D. latum* are mammals feeding on fish, such as dogs, cats, foxes, seals, and others. Infected mammals excrete cestode eggs, and in the aquatic environment, a ciliated coracidium hatches from the eggs. Coracidia are consumed by copepods (e.g., Cyclops and Diaptomus), which function as the first intermediate host. In the copepod’s body cavity the oncosphere develops into the next larval form—the procercoid. Copepods with the procercoids are eaten by fish, which are the second intermediate host. They can be freshwater fish, such as pike, perch, salmon, sea trout, eels, and fish inhabiting brackish water. In the body of fish, the procercoid develops into the next larval stage—the plerocercoid. The location of the plerocercoid in the fish may vary: the body cavity, liver, spleen, gonads, or muscles. The plerocercoid can attain a length of about 30 mm and has a cephalic furrow. When the definitive host ingests the plerocercoid together with the fish, the cestode matures into the adult, sexually mature form [60,61].

Another parasite that can pose a threat to humans is the nematode *Anisakis simplex*, recorded in tub gurnard, Atlantic mackerel, and dab. Typical definitive hosts of *A. simplex*, in which it develops and reaches sexual maturity, are dolphins, seals, porpoises (*Phocoena*), and whales in the seas and oceans of the northern hemisphere. The third-stage larvae (L3) of this nematode can infect people via fish, such as garfish, Baltic herring, lumpfish, Baltic cod *Gadus morhua callarias* L., 1758, three-spined stickleback *Gasterosteus aculeatus* L., 1758, ide *Leuciscus idus* (L., 1758), European flounder, and pike-perch *Sander lucioperca* (L., 1758) [46]. The geographic distribution of the *A. simplex* population around the world is varied and includes an Atlantic and a Pacific subpopulation [62]. This nematode is particularly common in the northern Atlantic Ocean [33]. *A. simplex* larvae are found in the Baltic in western herring, which come from the Danish straits and North Sea to spawn [33]. In Poland, it has been found in herring, Atlantic cod, garfish, and other fish [33]. Sobecka [43] found *A. simplex* in cod from most of the fishing grounds included in her study, i.e., the Barents Sea, the Irminger Sea, Øresund, the Gulf of Gdansk, and the Pomeranian Bay. Klimpel et al. [45] recorded this nematode in surmullet in the North Sea (prevalence 3.8%) [45]. Skrzypczak et al. [63] found small numbers of *A. simplex* (L3 larvae, 0.8%) in seals on the southern coast of the Baltic Sea. Fish entering the Baltic from the North Sea can transport parasites between these water bodies. This indicates that *A. simplex* can be transmitted in this area, especially from rare fish that are transient migrants. Although the scale of this phenomenon is minor as it involves fish that are rare in the Baltic, due to the importance of the nematode *A. simplex* for human health, it should be taken into account when considering the development of the population of these nematodes in the Baltic.

The presence of *A. simplex* in humans can cause a syndrome known as anisakiasis. Anisakiasis is most common in regions with a tradition of raw fish consumption. People become infected by eating fresh raw fish containing live third-stage larvae of *Anisakis*. After entering the human digestive tract, the larvae penetrate the mucosa and submucosa of the small intestine or stomach. Humans are an incidental host of *A. simplex*, so nematodes do not achieve maturity but die after a few weeks. Audicana et al. [64] draw attention to the occurrence of severe allergic reactions in some infected people, including hives and anaphylactic shock. Allergic reactions may be caused by both live and dead nematodes in human tissues [64]. Kołodziejczyk et al. [65] described the first case of human anisakiasis in Poland, caused by the living third-stage larva of *Anisakis simplex sensu stricto*, in a patient who had eaten raw Atlantic salmon (*Salmo salar* L., 1758). The parasite caused gastrointestinal symptoms [65].

*Pseudoterranova decipiens* was found in the pyloric caeca of tub gurnard. The geographic distribution of this nematode is Holarctic. Fish are the second and paratenic intermediate hosts. The first intermediate hosts are various benthic organisms, and the definitive hosts are various species of seal. In the Baltic, this nematode was previously recorded in Atlantic cod. Więcaszek et al. [13] found larvae (L3) of this nematode in the stomach of surmullet, on the pyloric caeca, and under the liver of tub gurnard. The incidental infection of humans with this nematode may cause anisakiasis. *Contracaecum osculatum* was found in tub gurnard and surmullet in the stomach and pyloric caeca. The larvae are found in fish of various species of Gadidae and Clupeidae. In Poland, it has been reported in Atlantic cod, herring, dab, lumpfish, and European flounder [33]. Seals are the definitive host [33]. The location of *C. osculatum* larvae (L3) and *P. decipiens* larvae (L3) in the digestive tract of fish is not typical. Fish are intermediate and paratenic hosts of these nematodes. *C. osculatum* larvae are most often located under the serous membrane of the liver and other organs of the body cavity of fish, while larvae of *P. decipiens* are found in the muscles of fish [33]. The presence of larvae in the digestive tract of the fish may indicate that infection had taken place recently and the larvae had not yet passed from the digestive system of the fish to their typical locations in the body.

The remaining nematodes detected in the fish were *Dichelyne* (*Cucullanellus*) *minutus*, *Raphidascaris acus,* and *Hysterothylacium aduncum L3/L4*. *Dichelyne* (*Cucullanellus*) *minutus* has previously been recorded in Poland in the Gulf of Gdansk [46], including in plaice. This nematode is distributed in the Atlantic Ocean, the Baltic Sea, the Black Sea, and the Pacific Ocean [66]. Adult nematodes settle in the intestine of fish, while nematode larvae may be present in the intestinal walls, liver, and peritoneum of other fish that are a component of their diet.

*Raphidascaris acus* was found in tub gurnard, in the intestine. This nematode with Palearctic geographic distribution is a parasite of the digestive tract of fish—mainly freshwater fish, and less often marine fish. Its larvae live in aquatic invertebrates and in the organs of the body cavity of fish (mainly cyprinids), which serve as intermediate and paratenic hosts [67]. It is a common parasite in Poland, noted in many species of freshwater fish in inland water bodies and the coastal waters of the Baltic. In an earlier study, Więcaszek et al. [14] reported its presence in the pyloric caeca of lumpfish *Cyclopterus lumpus*—the first record of the presence of *R. acus* in this host [14]. Sobecka [43] noted the presence of this nematode in cod from the Gulf of Gdansk.

*Hysterothylacium aduncum* was detected in surmullet. The geographic distribution of this nematode includes the Baltic Sea, the North Sea, the White Sea, the Mediterranean Sea, and the Atlantic Ocean. Its definitive hosts are various species of predatory fish in freshwater and brackish water, including alice shad *Alosa alosa* (L.), garfish, and viviparous eelpout, which are considered to be the main hosts in the Baltic. The paratenic hosts are planktivorous fish, including Atlantic salmon, perch, European eel *Anguilla anguilla* (L. 1758), and others. In Poland, it has been recorded in numerous fish species. Its intermediate hosts are planktonic crustaceans *Acartia bifilosa* (Giesbrecht, 1881) and *Eurytemora affinis* (Poppe, 1880) [33]. A study by Klimpel et al. [45] on the distribution of parasites of red mullet in the North Sea and Mediterranean Sea showed the presence of *H. aduncum* in this fish at all locations studied—the North Sea, the Ligurian Sea, the Tyrrhenian Sea and the Adriatic Sea, with a prevalence ranging from 4.2% in the Adriatic Sea to 51.9% in the North Sea [45]. Sobecka [43] reported the occurrence of this nematode in cod from most of the fishing grounds included in the study (the Barents Sea, the Irminger Sea, Øresund, the Gulf of Gdansk, and the Pomeranian Bay).

The acanthocephalan *Echinorhynchus gadi* was recorded in saithe and dab. It is one of the most widespread species in North European marine fish and is the predominant parasite in the Atlantic cod, which is the definitive host [16]. Amphipods are its intermediate host. The severe infection of this large cod (which normally does not feed on amphipods) may be explained by the transmission of parasites from prey fish to the large predatory cod. These prey fish may include young cod [58,68]. The presence of this acanthocephalan has been noted in cod from various fishing grounds, including the Pomeranian Bay [43].

### 4.2. Protozoans

*Ichthyophonus hoferi* was recorded in European hake, tub gurnard, Atlantic mackerel, and brill. It is a principally marine species and is widespread in coastal and open-ocean areas in the northern and southern hemispheres. *I. hoferi* attacks more than 80 marine and freshwater species of fish [69]. This infection causes the disease ichthyophoniasis, the consequences of which are difficult to assess for wild populations of fish. According to a number of authors, the infection of fish with *I. hoferi* occurs when the fish ingest food containing viable spores of the pathogen [70]. It is capable of horizontal transmission between fish in freshwater by contact or feeding on infected dead fish [71]. It has been studied in naturally infected Baltic herring, in sprat, and in flounder from the west coast of Sweden [72]. In a study of cod from various fishing grounds, *I. hoferi* was found only in cod from the Irminger Sea [43]. Repercussions of the disease for wild populations of fish are, as a rule, identifiable only after an epizootic [73].

*Trichodina jadranica* was noted sporadically on the gills of brill. Protozoa of the genus *Trichodina* are widespread in freshwater and marine fish. *T. jadranica* is found in marine environments in various species of fish and is a typical parasite of European flounder [74,75] and European eel [76]. It can cause health problems in fish in aquaculture, e.g., [77]. *T. jadranica* was first recorded in brill by Więcaszek et al. [15]; this was also the first record of this parasite in the Pomeranian Bay [15].

### 4.3. Intestinal Contents

The diet of the fish examined in our study contained the invertebrates *Crangon crangon*, Mysidacea and *Mytilus* sp., as well as the remains of fish: freshwater tench, European flounder, and round goby. Most of the fish specimens had empty stomachs (68.4%), which may have been due to their new environment, especially if they were translocated with saltwater inflows from the North Sea within a short time. Another cause may be a narrow food spectrum or unwillingness to eat novel prey. To date, little attention has been given to predator naivety, which can result in low predation pressure on the novel species in a new environment [78].

Due to the specific diet structures of the fish species examined in the study, it is possible that the fish were not food generalists and could not survive in the Baltic Sea. For example, the European hake is not an opportunistic predator [79], and the stomach of the specimen caught in this study was empty. However, the remains of *Bylgides sarsi* were reported in the gut of a specimen from Lithuania [80]. In the stomach of the plaice, highly digested remains of *Mytilus* sp. were noted. A molecular study of the additional sample of *Mytilus* sp. from the seafloor showed the presence of *M. edulis*. It is noteworthy that *M. edulis* colonizes the western Baltic (e.g., the Kattegat and the Belt Sea) at salinities between 12 and 25, while *Mytilus trossulus* is recorded in the eastern Baltic Proper at salinities between 4.5 and 8 [81]. *Mytilus edulis* was reported in the stomachs of plaice and dab from the Kattegat [82]. Molluscs and gastropods were the main prey for both species from this area. We also detected new food items in the diet of fish. Round goby, which has become an important food item for piscivorous fish in the Pomeranian Bay [83], was found for the first time in the stomach of one specimen of the common sole. One round goby specimen was also noted in the stomach of the brill, but this was first recorded in 2014 [15]. The freshwater tench was noted for the first time in the diet of saithe. The occurrence of European flounder in the stomach of a tub gurnard was also reported for the first time in the diet of this species [84]. Crustaceans, molluscs, and fish often function as intermediate hosts of parasites in the environment. Many fish species are known as paratenic hosts of parasites. Our study is limited by the migratory nature of the fish and the fact that we could not compare the feeding behaviour of fish in the study area with that of fish outside the study area.

## 5. Conclusions

An important discovery is the presence of the nematode *Capillaria* (*Procapillaria*) *gracilis* (in saithe), a typical marine parasite rarely recorded in Poland. An important discovery for public health is the presence of parasites known to pose a risk to human health (*Anisakis simplex*, *Pseudoterranova decipiens*, and *Contracaecum osculatum*) in rare fish species in the Baltic. In the case of *Diphyllobothrium* sp., we cannot estimate the risk of infection in humans.

Because the fish species examined in the study do not typically occur in the Baltic and information on how long they feed while in the Baltic is lacking, it cannot be conclusively stated which parasites were acquired by fish in the Baltic and coastal waters and which were introduced during migration. Most of the fish had empty stomachs and it cannot be stated conclusively whether they had eaten any food at all at the site where they were caught. Despite the fact that these fish are outside of their normal range, there are enough suitable intermediate hosts in the new habitat for the parasites to be able to complete their life cycles and survive in the ecosystem. A future direction of research should be monitoring rare fish species, especially commercial species, and their parasites.

## Figures and Tables

**Figure 1 animals-14-01029-f001:**
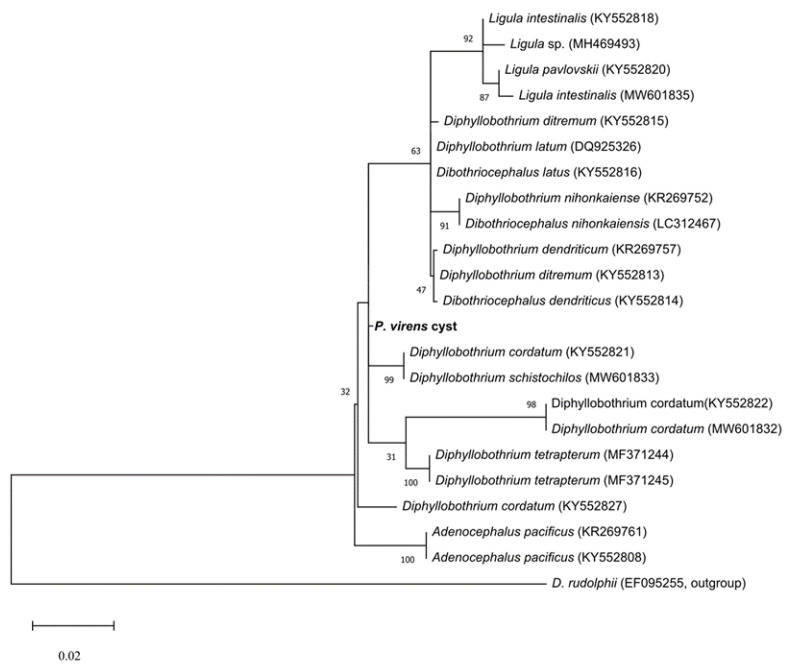
Maximum likelihood (ML) tree showing the phylogenetic relationships between haplotypes of the cyst found in *P. virens* and other species of the family Diphyllobothriidae. Values near branches show bootstrap support (BS). The maximum likelihood was computed using the Tamura–Nei substitution model in MEGA X, https://www.megasoftware.net. The scale bar indicates the number of substitutions per sequence position.

**Table 1 animals-14-01029-t001:** Study material, fish from the Pomeranian Bay and the Szczecin Lagoon (2010–2019).

Fish Species	Number of Fish Examined	Area of Catch	Fishing Period	Occurrence in Area of Catch
*Pollachius virens*, saithe	1	Szczecin Lagoon	late summer	very rare in the Pomeranian Bay; first record in the Szczecin Lagoon [4]
*Merluccius merluccius*, European hake	1	Pomeranian Bay	autumn	first record [4]
*Limanda limanda*, dab	2	Pomeranian Bay	summer	rare
*Solea solea*, common sole	2	Pomeranian Bay	spring	first record [4]
*Merlangius merlangus*, whiting	1	Pomeranian Bay	autumn	rare
*Chelidonichthys lucerna*, tub gurnard	5	Pomeranian Bay	autumn/winter	very rare
*Mullus surmuletus*, surmullet	2	Pomeranian Bay	autumn	very rare
*Scomber scombrus*, Atlantic mackerel	3	Pomeranian Bay	spring	rare
*Scophthalmus rhombus*, brill	2	Pomeranian Bay	spring and autumn	very rare
*Pleuronectes platessa*, plaice	1	Pomeranian Bay	autumn	rare

**Table 2 animals-14-01029-t002:** List of species/taxa of parasites of fish from the Szczecin Lagoon and Pomeranian Bay (2010–2019).

Parasite Taxon	Host	Intensity of Infection	Location in Host
PROTOZOA			
*Ichthyophonus hoferi*	*Merluccius merluccius*	numerous	gonads
	*Chelidonichthys lucerna*	numerous	gallbladder, urinary bladder
	*Scomber scombrus*	occasional	gonads
	*Scophthalmus rhombus*	occasional	kidney
*Trichodina jadranica*	*Scophthalmus rhombus*	occasional	gills
CESTODA			
*Diphyllobothrium* sp., plerocercoid	*Pollachius virens*	2	in the stomach *
NEMATODA			
*Dichelyne* (*Cucullanellus*) *minutus*	*Pleuronectes platessa*	71	intestine
*Capillaria* (*Procapillaria*) *gracilis*	*Pollachius virens*	2	intestine
*Raphidascaris acus*	*Chelidonichthys lucerna*	1	intestine
*Anisakis simplex*, larva (L3)	*Chelidonichthys lucerna*	1	liver
	*Scomber scombrus*	2	peritoneum, gonads
	*Limanda limanda*	3	intestine, gonads
*Contracaecum osculatum*, larva (L3)	*Chelidonichthys lucerna*	9	on the pyloric caeca *, in the stomach *
	*Mullus surmuletus*	2	intestine
*Hysterothylacium aduncum*, larva (L3/L4)	*Mullus surmuletus*	2	intestine
*Pseudoterranova decipiens*, larva (L3)	*Chelidonichthys lucerna*	3	on the pyloric caeca *
ACANTHOCEPHALA			
*Echinorhynchus gadi*	*Pollachius virens*	2	intestine
	*Limanda limanda*	3	intestine

* atypical location in the host.

**Table 3 animals-14-01029-t003:** Stomach contents of fish specimens (numbers of fish examined are in brackets).

Species	Stomach Content (Number of Fish Specimens in Brackets)
*Merlangius merlangus*	(1) one specimen of *Crangon crangon* (Decapoda, Malacostraca)
*Pollachius virens*	(1) scales of *Tinca tinca* (Cypriniformes)
*Merluccius merluccius*	(1) empty stomach
*Chelidonichthys lucerna*	(1) remains of *C. crangon* (Decapoda, Malacostraca); three skeletons of *Platichthys flesus* (Pleuronectiformes) (1) remains of Mysidacea (Mysida, Malacostraca) (3) empty stomachs
*Mullus surmuletus*	(2) empty stomachs
*Scomber scombrus*	(3) empty stomachs
*Scophthalmus rhombus*	(1) one specimen of round goby *Neogobius melanostomus*, 2.9 cm TL (Gobiiformes) (1) empty stomach
*Pleuronectes platessa*	(1) *Mytilus* sp. (Bivalvia, Mytilida) (approx. 30 specimens, ingested remains), fish roe
*Limanda limanda*	(2) empty stomachs
*Solea solea*	(1) one specimen of round goby *N. melanostomus*, 4.1 cm TL (Actinopterygii, Gobiiformes) (1) empty stomach

## Data Availability

The research data used and analysed during this study are available from the authors (I.R. and B.W.) upon reasonable request.

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
