# Peer review of "Diphyllobothrium sp. and Other Parasites of Migrating and Rare Fish Species in the Southern Baltic Sea and Coastal Waters, Poland"

_animals, 2024, doi:10.3390/ani14071029_

Round 1
Reviewer 1 Report
Comments and Suggestions for Authors
Review report
Title: Diphyllobothrium sp. and other parasites of migrating and rarefish species in the Southern Baltic Sea and coastal waters, land
MS id: animals-2884551
The study addresses a relevant topic: identifying parasites in commercially important fish species and understanding their potential impact on human health. The use of a variety of fish species and a ten-year timeframe adds robustness to the data collection. The identification of a new parasite species (Capillaria gracilis) for Poland is a valuable contribution. However before it may reconsider for further processing following correction are recommended as major revisions.
· Abstract:
o Clarity: The abstract could benefit from clearer and more concise language. Avoid jargon like "migrating and rare fish" and replace it with specific species names or categories (e.g., "commercially important fish not native to the Baltic Sea").
o Focus: The abstract should prioritize the key findings and highlight the new parasite record. Minimize details like the specific methods used.
· Introduction:
o Context: Provide stronger context regarding the potential risks associated with zoonotic parasites (parasites transmissible between animals and humans) in commercially fished species.
o Knowledge gap: Clearly state the existing knowledge gap around parasite acquisition during fish migration and its implications for consumer safety and ecosystem health.
· Methods:
o Justification: Briefly explain the rationale for selecting the specific fish species. Was there a specific hypothesis regarding their parasite composition?
o Sample size: Mention the number of fish analyzed for each species. Was the sample size sufficient to draw statistically significant conclusions?
· Results:
o Empty stomachs: Expand on the observation of mostly empty stomachs. Discuss potential reasons (e.g., seasonal variations, capture methods) and how it impacted the ability to analyze food composition.
o Parasite discussion:
§ Differentiate between native and non-native parasite species found.
§ Emphasize the potential public health significance of zoonotic parasites.
§ Discuss the limitations of the study due to the migratory nature of the fish and the lack of information on feeding behavior outside the study area.
· Discussion:
o Implications:
§ Discuss the broader ecological implications of the identified parasites, considering potential impacts on other fish populations and the ecosystem.
§ Highlight the importance of further research on parasite dynamics in migratory fish and suggest future research directions.
Comments on the Quality of English Languagea through revision is recommended for,language and grammar
Author Response
We have revised the manuscript in accordance with all of the reviewers’ comments.
We agree with all of the reviewer’s comments. The authors’ responses to each comment are presented below.
Review report
Title: Diphyllobothrium sp. and other parasites of migrating and rarefish species in the Southern Baltic Sea and coastal waters, land
MS id: animals-2884551
The study addresses a relevant topic: identifying parasites in commercially important fish species and understanding their potential impact on human health. The use of a variety of fish species and a ten-year timeframe adds robustness to the data collection. The identification of a new parasite species (Capillaria gracilis) for Poland is a valuable contribution. However before it may reconsider for further processing following correction are recommended as major revisions.
- Thank you for appreciating the value of our research.
- We had written in the paper that C. gracilis is a new species in Poland, but in the course of our revision following review, we found literature data indicating that it had been recorded in Poland prior to our research. We have corrected this error in the manuscript and written that it is a rare species in Poland.
- Abstract:
o Clarity: The abstract could benefit from clearer and more concise language. Avoid jargon like "migrating and rare fish" and replace it with specific species names or categories (e.g., "commercially important fish not native to the Baltic Sea").
- Thank you for this constructive comment.
"Commercially important fish not native to the Baltic Sea" would not be an accurate description. Some of the fish species are native but very rare in the region, while others are non-native and were recorded here for the first time.
"Migrating fish” and “rare fish" are common expressions used in fish biology and ecology.
Examples:
Mincarone, M. M., Eduardo, L. N., Di Dario, F., Frédou, T., Bertrand, A., & Lucena-Frédou, F. (2022). New records of rare deep-sea fishes (Teleostei) collected from off north-eastern Brazil, including seamounts and islands of the Fernando de Noronha Ridge. Journal of Fish Biology, 101(4), 945–959. https://doi.org/10.1111/jfb.15155
Tamario et al. 2019. Ecological and Evolutionary Consequences of Environmental Change and Management Actions for Migrating Fish. Front. Ecol. Evol., Sec. Behavioral and Evolutionary Ecology; Vol. 7 - 2019 | https://doi.org/10.3389/fevo.2019.00271
o Focus: The abstract should prioritize the key findings and highlight the new parasite record. Minimize details like the specific methods used.
- We agree and have revised the Abstract.
- Introduction:
o Context: Provide stronger context regarding the potential risks associated with zoonotic parasites (parasites transmissible between animals and humans) in commercially fished species.
- Thank you for this constructive comment; we have added some information to the Introduction, but we discuss the question of the risk caused by fish parasites at greater length in the Discussion.
o Knowledge gap: Clearly state the existing knowledge gap around parasite acquisition during fish migration and its implications for consumer safety and ecosystem health.
- We have added information about the knowledge gap in the Introduction.
- Methods:
o Justification: Briefly explain the rationale for selecting the specific fish species. Was there a specific hypothesis regarding their parasite composition?
- We have added appropriate information in the Materials and Methods section.
The fish species from which parasites were collected for analysis are very rare in the Pomeranian Bay. They had probably migrated from the North Sea. During monitoring studies of commercial species in the Bay (2010-2019), single specimens of very rare species which had appeared during fishing were set aside for biological, genetic and parasitological research. The presence of European hake, common sole, and dab was recorded for the first time in the Pomeranian Bay (Więcaszek et al. 2023).
We hypothesized that these fish may transport parasite species which can pose a threat to native species. Another interesting question was whether we would find parasites known in fish in the Pomeranian Bay but previously unknown in these alien fish species.
The results of biological and genetic analyses of fish (20 specimens) is presented in a study by Więcaszek B., Panicz R., Eljasik P. , Tański A., Bushra A. 2023. Molecular and biological studies of nonindigenous and extremely rare fish species from the western Baltic reported from the Pomeranian Bay (southwest Baltic Proper). Folia Biologica, vol. 71 No 4: 181-194.
o Sample size: Mention the number of fish analyzed for each species. Was the sample size sufficient to draw statistically significant conclusions?
- The number of fish analysed is presented in Table 1. In our opinion, 20 specimens belonging to 10 species is insufficient to draw statistically significant conclusions.
- Results:
o Empty stomachs: Expand on the observation of mostly empty stomachs. Discuss potential reasons (e.g., seasonal variations, capture methods) and how it impacted the ability to analyze food composition.
- The stomachs of the fish were often empty, which may have been due to their new environment, especially if they were translocated with saltwater inflows from the North Sea within a short time. Fish may have a narrow food spectrum or exhibit naivety towards novel prey species. To date, little attention has been given to predator naivety, which can result in low predation pressure on the novel species in the new environment (Reid et al. 2010).
Reid, Amelia & Seebacher, Frank & Ward, Ashley. (2010). Learning to hunt: The role of experience in predator success. Behaviour. 147. 223-233. 10.1163/000579509X12512871386137
o Parasite discussion:
- Differentiate between native and non-native parasite species found.
- Emphasize the potential public health significance of zoonotic parasites.
- Discuss the limitations of the study due to the migratory nature of the fish and the lack of information on feeding behavior outside the study area.
- We have added appropriate information in the text. Capillaria (Procapillaria) gracilis is a species whose occurrence in the Baltic is not typical, probably due to low salinity. Thus it can be considered a non-native species. The remaining parasite species found in the fish could potentially complete their life cycle in the Baltic.
- We have added some new information to the manuscript to better focus on public health, as suggested by all of the reviewers.
- Our study is limited by the migratory nature of the fish and the fact that we could not compare the feeding behaviour of fish in the study area with that of fish outside the study area. We checked the diet composition of the fish species in the literature to determine whether we had found any new components in the stomachs of fish in the Pomeranian Bay. We added information about new diet components: round goby is a new diet component in common sole, and the freshwater species tench is a new diet component in the saithe.
The source materials regarding the diet composition of the fish included the following:
Froese and Pauly [Fishbase] (2023), Heesen et al. 2015, Więcaszek et al. 2019, Marie Henriette Du Buit, 1996 (Diet of hake (Merluccius merluccius) in the Celtic Sea, Fisheries Research,Volume 28, Issue 4: 381-394),
Pavicic et al. 2018 (Feeding habits of the striped red mullet, Mullus surmuletus in the eastern Adriatic Sea. Acta Adriatica, Vol. 29: 123-1260);
Kvaavik et al. 2019 (Diet and feeding strategy of Northeast Atlantic mackerel (Scombrus scomber) in Icelandic waters. PLoS One. 2019 Dec 30;14(12):e0225552. doi: 10.1371/journal.pone.0225552);
Tyrrell et al. 2007. (The dynamic role of pollock (Pollachius virens) as a predator in the Northeast US continental shelf ecosystem: a multi-decadal perspective. J. Northw. Atl. Fish. Sci., 38: 53–65. https://doi.org/10.2960/J.v38.m605), Fanelli et al. 2022. (Seasonal Trophic Ecology and Diet Shift in the Common Sole Solea solea in the Central Adriatic Sea. Animals 2022, 12, 3369. https://doi.org/10.3390/ani12233369)
- Discussion:
o Implications:
- Discuss the broader ecological implications of the identified parasites, considering potential impacts on other fish populations and the ecosystem.
- Highlight the importance of further research on parasite dynamics in migratory fish and suggest future research directions.
- We have added new information to the manuscript. Because the fish analysed are rare in the Baltic and our sample of fish was small, it is difficult to assess their impact on other populations and the ecosystem. On the one hand, parasites are a natural component of every ecosystem (PojmaÅ„ska 2002). On the other hand, migrating fish can contribute to the translocation of alien parasite species between ecosystems, which potentially can influence the qualitative and quantitative parameters of the occurrence of parasites in native fish species and other organisms which can act as their hosts (MoroziÅ„ska-Gogol 2015). Another question is the impact on human health – the absence of parasites does not always mean a ‘healthy’ ecosystem, as the qualitative and quantitative composition of parasites depends on multiple factors (e.g. the presence of suitable hosts, salinity, temperature, or pollution of the environment). This information should be taken into account in implementing the One Health concept (Evans B.R., Leighton F.A.: A history of One Health. Rev. sci. tech. Off. int. Epiz. 2014, 33, 413–420).
- A future direction of research should be monitoring of rare fish species, especially commercial species, and their parasites.
We have also reduced the number of subsections in the Discussion section for the sake of clarity.
Reviewer 2 Report
Comments and Suggestions for Authors
Reviewer’s comments
This is a mainly well-written and interesting work identifying parasites from fish that are unusual to Polish waters. The subject is relevant to the issue of global climate change which is altering the range of marine fish species and their parasites. Although host sample sizes were small, the authors’ investigation of each host appears to have been thorough. I would have been more confident in the identifications of the parasites, if DNA sequence had been obtained for all of the parasites.
It is not clear why the authors chose to sequence the DNA of just Diphyllobothrius sp. and the mussel found as a food item. Ideally, the authors did would have sequenced each of the parasites (and food items?) found. This lack of DNA evidence is a shortcoming of the study, so a sentence explaining why only these two taxa were sequenced is necessary.
Line 18 “acanthocephalan” singular
Table 1
Mullus surmuletus (L. 1758) should not have parentheses. Linnaeus described the species under this genus and it hasn’t been moved to another genus.
Scomber scombrus (L., 1758) should not have parentheses. Linnaeus described the species under this genus and it hasn’t been moved to another genus.
Scopthhalmus rhombus should be Scophthalmus rhombus
Pleuronectes platessa (L., 1758) should not have parentheses. Linnaeus described the species under this genus and it hasn’t been moved to another genus.
Line 173 and table 2.
It is strange that Diphyllobothrius plerocercoids were found encysted in the stomach of the saithe. Whether the saithe was a second intermediate or paratenic host, it is expected that the larval cestode would be found in the body cavity or tissues, as mentioned in lines 267-8. Maybe this anomaly is worth mentioning?
Line 174 and elsewhere.
While we understand that the taxonomy of the diphyllobothriids is open to argument, could the authors settle on either Diphyllobothrium or Dibothriocephalus as the genus found in the saithe? It feels lacking in taxonomic confidence to call the cestode by both alternative names! A very simple decider may be in Scholz, Kuchta and Brabec 2019 (IJP: Parasites and Wildlife 9L 359-369) who place terrestrial species in Dibothriocephalus and marine species in Diphyllobothrium, makingDiphyllobothrium the name to use here. If both names are still presented as alternatives, the reason for this should be stated.
Line 189.
While it can be useful to note synonyms, consistency should dictate that all species mentioned should have all, or no, synonyms mentioned. For instance there are at least 5 synonyms for Dichelyne (C.) minutus, and more than 20 for H. aduncum, so the authors may prefer to leave out synonyms! If the synonym for D. (C.) minutus is relevant because it has been cited under that name in previous works, it can still be mentioned, but the work in which it was cited should be referenced.
Line 196
Neither Contracaecum osculatum nor Pseudoterranova decipiens mature in fish, so it is rare to find them in the digestive tract. If the authors are sure of their identifications, perhaps they could mention this anomaly? This is somewhere the reader would have more confidence in the identification if there were DNA sequences to back it up.
Line 197
Capillaria (Procapillaria) gracilis (Bellingham, 1940) should have its authority as it’s the first mention in the body of the text.
Line 261
“a coracidium with an oncosphere larva hatches from the eggs “ – Not quite correct. The oncosphere develops into the coracidium inside the egg, so it is sufficient to say “a ciliated coracidium hatches from the eggs”.
Lines 298-300
Italics for species names
Line 312-3
Chironomid larvae are alternative intermediate hosts, not paratenic hosts of Capillaria gracilis.
Line 390
I disagree that new food-web connections were made. Records of foods and those of gastrointestinal parasites cannot be conclusively stated to be trophic links unless they are directly connected by 1) finding the parasite within the prey item, within the predator, or 2) DNA sequencing links the parasite in the intermediate host with that in the definitive/paratenic host. I think the most that can be inferred is that, despite these fish being out of their normal ranges, there are sufficient suitable intermediate hosts in the new environment for their parasites to compleete their life cycles and persist in the population.
Author Response
We have revised the manuscript in accordance with all of the reviewers’ comments.
We agree with all of the reviewer’s comments. The authors’ responses to each comment are presented below.
Reviewer’s comments
This is a mainly well-written and interesting work identifying parasites from fish that are unusual to Polish waters. The subject is relevant to the issue of global climate change which is altering the range of marine fish species and their parasites. Although host sample sizes were small, the authors’ investigation of each host appears to have been thorough. I would have been more confident in the identifications of the parasites, if DNA sequence had been obtained for all of the parasites.
It is not clear why the authors chose to sequence the DNA of just Diphyllobothrius sp. and the mussel found as a food item. Ideally, the authors did would have sequenced each of the parasites (and food items?) found. This lack of DNA evidence is a shortcoming of the study, so a sentence explaining why only these two taxa were sequenced is necessary.
- We would like to thank the reviewer for the positive comments regarding our work.
- The DNA of Diphyllobothrium sp. was sequenced so that we could be certain about the identification of the larva. In the case of this tapeworm, molecular tools were needed to confirm its genetic identity. Identification of the other parasites on the basis of morphological characters was sufficient, so there was no need to carry out DNA testing. The fish prey in the stomachs was identified on the basis of morphological characters, using keys cited in the paper. There was no need for molecular analysis. The only exception was mussels of the genus Mytilus found in the stomach of plaice. Because they were quite well digested, genetic analysis indicated only the genus Mytilus. Mussels of this genus were collected as quickly as possible from the site where the plaice had been caught, and these were identified as M. edulis. It is true, however, that the M. edulis identified did not come from the stomach of the fish, so we can only surmise that it was M. edulis. We have revised the manuscript accordingly.
Line 18 “acanthocephalan” singular
- Corrected
Table 1
Mullus surmuletus (L. 1758) should not have parentheses. Linnaeus described the species under this genus and it hasn’t been moved to another genus.
Scomber scombrus (L., 1758) should not have parentheses. Linnaeus described the species under this genus and it hasn’t been moved to another genus.
Scopthhalmus rhombus should be Scophthalmus rhombus
Pleuronectes platessa (L., 1758) should not have parentheses. Linnaeus described the species under this genus and it hasn’t been moved to another genus.
- Corrected. We have written the author’s name and the year the species was first described the first time the species is mentioned in the text. We have removed this information thereafter, in the text and in the tables.
Line 173 and table 2.
It is strange that Diphyllobothrius plerocercoids were found encysted in the stomach of the saithe. Whether the saithe was a second intermediate or paratenic host, it is expected that the larval cestode would be found in the body cavity or tissues, as mentioned in lines 267-8. Maybe this anomaly is worth mentioning?
- We agree. We have added the information that this is an atypical location for the parasite in the host’s body in Table 2 and in the text, where we describe Diphyllobothrium sp. The saithe, the marine species in which we found Diphyllobothrium, was caught in the freshwater Szczecin Lagoon, which is connected to the Pomeranian Bay. In its stomach we found scales of the freshwater tench (Tinca tinca), which may have been the source of the tapeworm. According to the database at https://www.fishbase.se/summary/Pollachius-virens.html, “Smaller fish of Pollachius virens in inshore waters feed on small crustaceans (copepods, amphipods, euphausiids) and small fish, while larger fish prey predominantly upon fishes”. Since only the scales of tench were found in the saithe, it seems likely that the source of the Diphyllobothrium sp. larvae was a fish of another species that had previously been eaten and digested (this was the second intermediate host), and that the larvae were found before they could migrate from the stomach to the muscles of the saithe. Diphyllobotrium sp. has been found in the stomach of cod (Karasev A.B., Mitenev V.K.,Shulman B.S. 1996. Ecological peculiarities of the parasite fauna of cod and polloc in the vicinity of the Kislaya Inlet Tidal power plant, Wester Murman (The Barents Sea). Sarsia 80: 307-312.).
Line 174 and elsewhere.
While we understand that the taxonomy of the diphyllobothriids is open to argument, could the authors settle on either Diphyllobothrium or Dibothriocephalus as the genus found in the saithe? It feels lacking in taxonomic confidence to call the cestode by both alternative names! A very simple decider may be in Scholz, Kuchta and Brabec 2019 (IJP: Parasites and Wildlife 9L 359-369) who place terrestrial species in Dibothriocephalus and marine speciesin Diphyllobothrium, making Diphyllobothrium the name to use here. If both names are still presented as alternatives, the reason for this should be stated.
- We agree with the reviewer. We have left only the name Diphyllobothrium sp. and have explained the use of name Diphyllobothrium, based on the cited literature.
Line 189. ency should dictate that all species mentioned should have all, or no, synonyms mentioned. F or instance there are at least 5 synonyms for Dichelyne (C.) minutus, and more
While it can be useful to note synonyms, consist than 20 for H. aduncum, so the authors may prefer to leave out synonyms! If the synonym for D. (C.) minutus is relevant because it has been cited under that name in previous works, it can still be mentioned, but the work in which it was cited should be referenced.
- We agree. We have removed the synonym from the manuscript.
Line 196
Neither Contracaecum osculatum nor Pseudoterranova decipiens mature in fish, so it is rare to find them in the digestive tract. If the authors are sure of their identifications, perhaps they could mention this anomaly? This is somewhere the reader would have more confidence in the identification if there were DNA sequences to back it up.
- We agree with the reviewer. In Table 2 and in the text we have revised and clarified the information given about of the location of the parasites in the fish.
Line 197
Capillaria (Procapillaria) gracilis (Bellingham, 1940) should have its authority as it’s the first mention in the body of the text.
- Corrected. We had written in the paper that Capillaria (Procapillaria) gracilis is a new species in Poland, but in the course of our revision following review, we found literature data indicating that it had been recorded in Poland prior to our research (recorded as Capillaria gracilis). We have corrected this error in the manuscript and written that it is a rare species in Poland.
Line 261
“a coracidium with an oncosphere larva hatches from the eggs “ – Not quite correct. The oncosphere develops into the coracidium inside the egg, so it is sufficient to say “a ciliated coracidium hatches from the eggs”.
- We agree, and we have revised the sentence.
Lines 298-300
Italics for species names
- Corrected.
Line 312-3
Chironomid larvae are alternative intermediate hosts, not paratenic hosts of Capillaria gracilis.
- Corrected.
Line 390
I disagree that new food-web connections were made. Records of foods and those of gastrointestinal parasites cannot be conclusively stated to be trophic links unless they are directly connected by 1) finding the parasite within the prey item, within the predator, or 2) DNA sequencing links the parasite in the intermediate host with that in the definitive/paratenic host. I think the most that can be inferred is that, despite these fish being out of their normal ranges, there are sufficient suitable intermediate hosts in the new environment for their parasites to compleete their life cycles and persist in the population.
- We agree and have revised this part of the text.
We have also reduced the number of subsections in the Discussion section for the sake of clarity.
Reviewer 3 Report
Comments and Suggestions for Authors
Animals 2884551 review
This manuscript provides an account of parasites found in fish in the Southern Baltic Sea with an attempt to relate the infections to the diet of the fish sampled.
General comments:
With only a total of 20 fish specimens caught over a 10 year period, and a maximum number of 5 individual fish per species examined, you must be very careful with the generalisations made about the ecological dynamics of the parasites found in these fish. I believe that this type of data set is imperative to be published, especially for fish that may be on the edges of their normal distributions, to ensure we know as much as possible about fish parasites, but do not overstate your results. For example, saying the results here give “new relationships in the food-web structure” (Line 390) is an overstatement. Most of the fish had empty stomachs (Line 369) and you cannot definitively state if any of the diet items were ingested by the fish within the location captured (Line 370).
Be careful of making the statement of zoonotic risk of parasites without reference to the location of the parasites in the fish. For the three species of parasites you list as a zoonotic risk – Diphyllobothrium sp., Anisakis simplex and Pseudoterranova decipiens – all were found in parts of the fish not commonly, if at all, consumed (as per your discussion on Line 251 for Diphyllobothrium)– eg, stomach wall, peritoneum, intestine and pyloric glands. Anisakis simplex could be a risk for people who consume gonads, but that was in 2-3 of the cases only (from what I can determine from the data presented in Table 2). Also, are all species of Diphyllobothrium a potential zoonotic risk? You do discuss diphyllobothriids in general and state “several species…are known to be capable of infecting people” (Line 257) but do not actually state how many/all.
Specific comments:
Line 14, Line 194: replace “:” with “,”
Line 15: add “,” after “plaice”
Line 16: replace “,” with “;” to be consistent with the rest of the sentence
Line 18: acanthocephalan
Line 56-70: I would suggest naming the fish as 4 separate groups (you already have the groups but the first two are subgroups and could get confusing): Group 1, seasonal migrants; Group 2, periodic migrants; Group 3, northward expansion; Group 4, overflow species. However, you need to make sure the differences between Groups 1 and 2 are well defined; if they are not and there is overlap, then these fish should just be combined into one group. Also, you need to better define what you mean by “overflow” – are these fish migrating/distributing further south due to climate change?
Line 79-90: please be consistent in the presentation of fish names. You have already introduced surmullet (Line 67), so the common name only can be used here, for example. Same with the thicklip mullet (introduced Line 81, repeated Line 85).
Line 81: Not all readers of this paper will have a sound parasitological knowledge so you should make sure the group for each parasite is listed as well. It might be easier to provide this information in a Table?
Line 134: Why was only the cestode sequenced? You should present reasons as to why the others were not (eg, low number of specimens, size of specimens, need to keep specimens intact for morphological identification). Although identified morphologically, it is good practice to try to have molecular confirmation wherever possible, especially as the taxonomy of some of these groups is very complex and historically inaccurate.
Line 151: I would be careful of trying to base your identification of the gut bivalve on morphological similarities (which you do not specify) with “samples from the seafloor” near where the fish were captured (Line 155), especially as the mussel specimens were “highly digested” and the length of time the fish have been in the area is unknown. It is possible that these fish may have ingested these mussels elsewhere, with digestion of the hard-shelled mussels taking a period of time, as the fish moved to the location where they were captured. It is most likely the specimens in the fish stomach were M. edulis, but without molecular confirmation from those specific specimens, you cannot really say that. I would personally prefer it if you left the identification as Mytilus sp., with a comment that they could be M. edulis due to the presence of this species in the area but this was not able to be confirmed due to the reasons you have highlighted.
Line 230-239: You discuss the importance of molecular confirmation of Diphyllobothrium infection, but do not discuss how your specimen places with the other species, especially those that are known to cause human infections.
Line 248: Please be consistent in the presentation of scientific names – here you have provided authorities for these parasites, but have not done this elsewhere.
Line 279: Klimpel et al. found Anisakis simplex larvae in fish in an area close to where your study was conducted. You make a point about how fish move into the Baltic Sea from other areas, including the North Sea (Line 58), but do not make that connection here. Highlight that this shows the possibility of transmission of parasites like Anisakis simplex in this area, especially from these rare fish that are transient migrants to the area that people might not immediately consider a risk.
Line 298: italicise the species names
Line 304: are the larval stages found in the same fish or in fish within the diet?
Line 350: In what ways is this a dangerous disease? Especially if the impacts on wild populations are difficult to assess? Be careful of making such statements without evidence/proof/data/details.
Line 390: Due to the points raised above about this, I would delete the sentence relating to “food-web structure”.
Line 396: You need to include the Author Contribution statement relative to your manuscript and the authors involved.
Line 406: You need to include the Institutional Review Board Statement statement relative to your manuscript.
Line 416: As this paper does not involve humans, this section can be deleted.
Line 423: You need to include the Data Availability statement relative to your manuscript.
Line 434: You need to include the Conflicts of Interest statement relative to your manuscript.
Line 477, Line 589: Italicise species names
Line 497: Be consistent with the presentation of article titles – here every word is capitalised, whereas the reference on Line 501 is written in sentence style.
Line 557: Be consistent with the presentation of the doi information – here it is not underlined, but elsewhere it is
Line 574, Line 579, Line 591: insert space between the two references
Comments on the Quality of English LanguageOverall, the quality of the English is very good but there are a few minor grammatical issues.
Author Response
We have revised the manuscript in accordance with all of the reviewers’ comments.
We agree with all of the reviewer’s comments. The authors’ responses to each comment are presented below.
Animals 2884551 review
This manuscript provides an account of parasites found in fish in the Southern Baltic Sea with an attempt to relate the infections to the diet of the fish sampled.
General comments:
With only a total of 20 fish specimens caught over a 10 year period, and a maximum number of 5 individual fish per species examined, you must be very careful with the generalisations made about the ecological dynamics of the parasites found in these fish. I believe that this type of data set is imperative to be published, especially for fish that may be on the edges of their normal distributions, to ensure we know as much as possible about fish parasites, but do not overstate your results. For example, saying the results here give “new relationships in the food-web structure” (Line 390) is an overstatement. Most of the fish had empty stomachs (Line 369) and you cannot definitively state if any of the diet items were ingested by the fish within the location captured (Line 370).
- We agree with the reviewer. We have deleted the sentence indicated by the reviewer (line 390) and revised the Conclusions. The fish the parasites were found in are very rare in the Pomeranian Bay, in many cases found for the first time in this region (European hake, common sole, and dab). Their appearance is probably linked to the extension of the species’ range due to climate warming (surmullet and tub gurnard) and to inflows of saltwater from the North Sea (WiÄ™caszek B., Panicz R., Eljasik P. , TaÅ„ski A., Bushra A. 2023. Molecular and biological studies of nonindigenous and extremely rare fish species from the western Baltic reported from the Pomeranian Bay (southwest Baltic Proper). Folia Biologica, vol. 71 No 4: 181-194. https://doi.org/10.3409/fb_71-4.18 e-ISSN 1734-9168).
The eastern boundary of the distribution for all examined species is Kattegat-Skagerrak, except for sole and plaice (ICES Division III SD 20-24), dab, and brill (SD 22-32) (ICES 2015; 2020; 2022). However, literature data suggest that these species are very rare in the Baltic Sea (Więcaszek et al., 2023).
The stomachs of the fish were often empty, which may have been due to their new environment, especially if they were translocated with saltwater inflows from the North Sea within a short time. Fish may have a narrow food spectrum or exhibit naivety towards novel prey species. To date, little attention has been given to predator naivety, which can result in low predation pressure on the novel species in the new environment. (Reid et al. 2010).
Reid, Amelia & Seebacher, Frank & Ward, Ashley. (2010). Learning to hunt: The role of experience in predator success. Behaviour. 147. 223-233. 10.1163/000579509X12512871386137
Be careful of making the statement of zoonotic risk of parasites without reference to the location of the parasites in the fish. For the three species of parasites you list as a zoonotic risk – Diphyllobothrium sp., Anisakis simplex and Pseudoterranova decipiens – all were found in parts of the fish not commonly, if at all, consumed (as per your discussion on Line 251 for Diphyllobothrium)– eg, stomach wall, peritoneum, intestine and pyloric glands. Anisakis simplex could be a risk for people who consume gonads, but that was in 2-3 of the cases only (from what I can determine from the data presented in Table 2). Also, are all species of Diphyllobothrium a potential zoonotic risk? You do discuss diphyllobothriids in general and state “several species…are known to be capable of infecting people” (Line 257) but do not actually state how many/all.
- We agree with the reviewer. We have added information regarding assessment of the risk of parasites to the Discussion. We have also added new information to the description of Diphyllobothrium sp.
Specific comments:
Line 14, Line 194: replace “:” with “,”
Line 15: add “,” after “plaice”
Line 16: replace “,” with “;” to be consistent with the rest of the sentence
Line 18: acanthocephalan
- Corrected.
Line 56-70: I would suggest naming the fish as 4 separate groups (you already have the groups but the first two are subgroups and could get confusing): Group 1, seasonal migrants; Group 2, periodic migrants; Group 3, northward expansion; Group 4, overflow species. However, you need to make sure the differences between Groups 1 and 2 are well defined; if they are not and there is overlap, then these fish should just be combined into one group. Also, you need to better define what you mean by “overflow” – are these fish migrating/distributing further south due to climate change?
- As suggested, we have added a description of the groups of fish to the Discussion.
- Several groups of fish species from which parasites were collected for analysis can be distinguished. The first group comprises occasional marine visitors that seasonally migrate to the Baltic from the open Northeast Atlantic, the North Sea, and the Kattegat but are unable to establish self-sustaining populations in the Baltic, such as tub gurnard. They are sporadically distributed in the western Baltic Sea. Another example is the mackerel, a species with long migratory patterns, whose migration to and from the Baltic is influenced in part by climatic variables, including temperature. Mackerel is regularly found only in the southern Baltic and the Kattegat, but spawns outside the Baltic Sea (HELCOM 2021). A second group of fish species is anticipated to expand its distribution northward in response to climate warming. The surmullet belongs to this group (Engelhard et al. 2011). The third group of species typically comprises fish that overflow from the Kattegat and the Belt Sea, including European hake, saithe and whiting, abundant in the northern North Sea, the Skagerrak, and the Kattegat, but absent from western part of the Baltic Sea (Heessen et al. 2015).
- Fish that “overflow” – this refers to passive translocation with inflows of waters from the North Sea.
-The remaining fish species are considered native to the Baltic, but not to the Pomeranian Bay (except for plaice). According to Heessen et al. (2015), brill is regularly observed off Cape Arkona and around Bornholm; however, there were no landings in subdivisions SD 24-32 in the years 2012–2016. The presence of brill specimens in the Pomeranian Bay could result from either active migration from these areas or passive translocation with inflows of waters from the western Baltic. The first record of brill in the Pomeranian Bay was in 2014 (WiÄ™caszek et al. 2019). Sole, as per Heessen et al. (2015), is a southern species commonly found in the Kattegat (salinity >25), seldom encountered in the Baltic, and it becomes increasingly scarce further south and northeast. Dab currently resides in the Kattegat and the western Baltic, whereas in the central Baltic the stock experienced a collapse after the Second World War and never recovered due to unfavourable hydrographic conditions and predation by cod (Ojaveer et al. 2010).
- Sole, dab and hake were recorded in the Pomeranian Bay for the first time. The remaining species had been recorded in this area very rarely over the last 20 years.
Line 79-90: please be consistent in the presentation of fish names. You have already introduced surmullet (Line 67), so the common name only can be used here, for example. Same with the thicklip mullet (introduced Line 81, repeated Line 85).
- We agree. We have used the common name and Latin name when we first mention the species, and the common name thereafter.
Line 81: Not all readers of this paper will have a sound parasitological knowledge so you should make sure the group for each parasite is listed as well. It might be easier to provide this information in a Table?
- We have added the name of the group of each parasite, as suggested. We decided to include this information in the text and not to create a new table.
Line 134: Why was only the cestode sequenced? You should present reasons as to why the others were not (eg, low number of specimens, size of specimens, need to keep specimens intact for morphological identification). Although identified morphologically, it is good practice to try to have molecular confirmation wherever possible, especially as the taxonomy of some of these groups is very complex and historically inaccurate.
- In the case of Diphyllobothrium, molecular tools were needed to confirm the genetic identity of the samples. The other parasites were identified on the basis of morphological characters, and morphological identification was sufficient. We have clarified this in the Materials and Methods.
Line 151: I would be careful of trying to base your identification of the gut bivalve on morphological similarities (which you do not specify) with “samples from the seafloor” near where the fish were captured (Line 155), especially as the mussel specimens were “highly digested” and the length of time the fish have been in the area is unknown. It is possible that these fish may have ingested these mussels elsewhere, with digestion of the hard-shelled mussels taking a period of time, as the fish moved to the location where they were captured. It is most likely the specimens in the fish stomach were M. edulis, but without molecular confirmation from those specific specimens, you cannot really say that. I would personally prefer it if you left the identification as Mytilus sp., with a comment that they could be M. edulis due to the presence of this species in the area but this was not able to be confirmed due to the reasons you have highlighted.
- The fish prey in the stomach was identified on the basis of morphological traits, according to the keys cited in the paper. There was no need for molecular testing. The only exception was mussels of the genus Mytilus found in the stomach of plaice. They were quite well digested, so genetic analysis indicated only the genus Mytilus. Mussels of this genus were collected as quickly as possible from the site where the plaice had been caught, and these were identified as M. edulis. It is true, however, that the M. edulis identified did not come from the stomach of the fish, so we can only surmise that it was M. edulis. We have revised the information in the Results and in Table 3 accordingly.
Line 230-239: You discuss the importance of molecular confirmation of Diphyllobothrium infection, but do not discuss how your specimen places with the other species, especially those that are known to cause human infections.
- We have expanded the discussion of Diphyllobothrium in the Discussion and added some information.
Line 248: Please be consistent in the presentation of scientific names – here you have provided authorities for these parasites, but have not done this elsewhere.
- Corrected.
Line 279: Klimpel et al. found Anisakis simplex larvae in fish in an area close to where your study was conducted. You make a point about how fish move into the Baltic Sea from other areas, including the North Sea (Line 58), but do not make that connection here. Highlight that this shows the possibility of transmission of parasites like Anisakis simplex in this area, especially from these rare fish that are transient migrants to the area that people might not immediately consider a risk.
- We have added information about this in our discussion of A. simplex in the Discussion section.
Line 298: italicise the species names
- Corrected
Line 304: are the larval stages found in the same fish or in fish within the diet?
- In fish within the diet; we have clarified this in the manuscript.
Line 350: In what ways is this a dangerous disease? Especially if the impacts on wild populations are difficult to assess? Be careful of making such statements without evidence/proof/data/details.
- We agree and have deleted the word ‘dangerous’.
Line 390: Due to the points raised above about this, I would delete the sentence relating to “food-web structure”.
- Deleted.
Line 396: You need to include the Author Contribution statement relative to your manuscript and the authors involved.
- We have added this statement.
Line 406: You need to include the Institutional Review Board Statement statement relative to your manuscript.
- There is no need for such a statement. The fish used in the study came from by-catch from commercial fishing in the Pomeranian Bay by fishermen. When they were pulled onto the deck, they were already dead. These specimens have also been used in other biological research. The results were published in a paper by WiÄ™caszek B., Panicz R., Eljasik P., TaÅ„ski A., Bushra A. 2023, ‘Molecular and biological studies of nonindigenous and extremely rare fish species from the western Baltic reported from the Pomeranian Bay (southwest Baltic Proper)’. Folia Biologica, vol. 71 No 4: 181-194. In that case, once the editors of the journal were informed of the source of the fish, no Ethics Committee approval was required.
Line 416: As this paper does not involve humans, this section can be deleted.
- We have deleted this section.
Line 423: You need to include the Data Availability statement relative to your manuscript.
- We have included this statement.
Line 434: You need to include the Conflicts of Interest statement relative to your manuscript.
- We have included this statement.
Line 477, Line 589: Italicise species names
- Corrected.
Line 497: Be consistent with the presentation of article titles – here every word is capitalised, whereas the reference on Line 501 is written in sentence style.
- Corrected.
Line 557: Be consistent with the presentation of the doi information – here it is not underlined, but elsewhere it is
- Corrected.
Line 574, Line 579, Line 591: insert space between the two references
- We have inserted a space.
We have also reduced the number of subsections in the Discussion section for the sake of clarity.
Round 2
Reviewer 1 Report
Comments and Suggestions for Authors
Article now can be accepted
Author Response
We would like to thank the reviewer for accepting our article.
Reviewer 3 Report
Comments and Suggestions for Authors
I thank the authors for undertaking their revisions. I am happy with the manuscript except for a few minor issues:
throughout the manuscript you refer to plyoric glands - should these be pyloric caecae?
Line 211 - spelling of Diphyllobothrium.
Line 233 - why is this not typical? (However, this sentence is more Discussion than Results, so should be moved).
Line 375-379 - if Dibthriocephalus is now the accepted name (Line 382), why is the parasite referred to as Diphyllobothrium throughout the manuscript?
Author Response
We have written our responses under the reviewer’s comments:
Comments and Suggestions for Authors
I thank the authors for undertaking their revisions. I am happy with the manuscript except for a few minor issues:
throughout the manuscript you refer to plyoric glands - should these be pyloric caecae?
- Corrected
Line 211 - spelling of Diphyllobothrium.
- Corrected
Line 233 - why is this not typical? (However, this sentence is more Discussion than Results, so should be moved).
- We have moved this sentence to the Discussion and added clarification: ‘The location of C. osculatum larvae (L3) and P. decipiens larvae (L3) in the digestive tract of fish is not typical. Fish are an intermediate and paratenic host of these nematodes. C. osculatum larvae are most often located under the serous membrane of the liver and other organs of the body cavity of fish, while larvae of P. decipiens are found in the muscles of fish [33]. The presence of larvae in the digestive tract of the fish may indicate that infection had taken place recently, and the larvae had not yet passed from the digestive system of the fish to their typical locations in the body.’
Line 375-379 - if Dibthriocephalus is now the accepted name (Line 382), why is the parasite referred to as Diphyllobothrium throughout the manuscript?
- Another reviewer suggested this change: „While we understand that the taxonomy of the diphyllobothriids is open to argument, could the authors settle on either Diphyllobothrium or Dibothriocephalus as the genus found in the saithe? It feels lacking in taxonomic confidence to call the cestode by both alternative names! A very simple decider may be in Scholz, Kuchta and Brabec 2019 (IJP: Parasites and Wildlife 9L 359-369) who place terrestrial species in Dibothriocephalus and marine species in Diphyllobothrium, makingDiphyllobothrium the name to use here. If both names are still presented as alternatives, the reason for this should be stated.”
- We have changed the sentence as follows: “For example, the cestode previously belonging to the genus Diphyllobothrium – Diphyllobothrium latum, with the currently accepted name of Dibothriocephalus latus (Linnaeus, 1758) [59] – is frequently mentioned in the literature.”